# *BRCA* Mutations and Breast Cancer Prevention

**DOI:** 10.3390/cancers10120524

**Published:** 2018-12-19

**Authors:** Joanne Kotsopoulos

**Affiliations:** 1Women’s College Research Institute, Women’s College Hospital, 76 Grenville Street, 6th Floor, Toronto, ON M5S 1B2, Canada; joanne.kotsopoulos@wchospital.ca; Tel.: +1-416-351-3732 (ext. 2126); 2Dalla Lana School of Public Health, University of Toronto, 155 College St, Toronto, ON M5T 3M7, Canada

**Keywords:** *BRCA*, breast cancer, prevention

## Abstract

Women who inherit a deleterious *BRCA1* or *BRCA2* mutation face substantially increased risks of developing breast cancer, which is estimated at 70%. Although annual screening with magnetic resonance imaging (MRI) and mammography promotes the earlier detection of the disease, the gold standard for the primary prevention of breast cancer remains bilateral mastectomy. In the current paper, I review the evidence regarding the management of healthy *BRCA* mutation carriers, including key risk factors and protective factors, and also discuss potential chemoprevention options. I also provide an overview of the key findings from the literature published to date, with a focus on data from studies that are well-powered, and preferably prospective in nature.

## 1. Overview of Breast Cancer Risk and Management among *BRCA* Mutation Carriers

Women who inherit a deleterious germline *BRCA1* or *BRCA2* mutation face high lifetime risks of developing breast cancer by age 80, which are estimated at 72% and 69%, respectively [1,2,3,4]. Once diagnosed with invasive breast cancer, these women have a high risk of developing a second ipsilateral [5] or contralateral breast cancer [6]. Women with an inherited mutation in either gene are also at a significantly elevated risk of developing ovarian cancer [1,7].

In a recent prospective analysis of 3886 *BRCA* mutation carriers, Kuchenbaecker et al. reported a cumulative incidence of breast cancer to age 70 of 66% for *BRCA1* and 61% for *BRCA2* mutation carriers [1]. The corresponding annual risk estimates were 2.18% and 1.98%, respectively. For *BRCA1* mutation carriers, the risk of breast cancer increased substantially between the ages of 30–50, while for women with a *BRCA2* mutation, the risks were highest between ages 40–60. The study by Kuchenbaecker et al. combined several population-based studies, which comprised an ethnically diverse population. We have recently estimated the risk of breast cancer among *BRCA1* mutation carriers among women of Polish decent who were carriers of one of three founder mutations [8]. The annual risk was 1.78% (with a maximum risk observed between ages 30–65), and the cumulative incidence of a primary breast cancer to age 70 was 52%. In the most recent prospective report, the cumulative risk of epithelial ovarian cancer to age 80 was 49% for *BRCA1* and 21% for *BRCA2* mutation carriers [7]. These risk estimates are in line with those of Kuchenbaecker et al., who reported a cumulative incidence of ovarian cancer of 44% for *BRCA1* and 17% for *BRCA2* mutation carriers [1].

*BRCA1*-associated breast cancers exhibit the pathological features of an aggressive phenotype, and are usually hormone receptor-negative, whereas *BRCA2*-assocated breast cancers tend to resemble sporadic cancers, and are predominantly hormone receptor-positive [9,10,11,12,13,14]. A study compared the survival experience of *BRCA2* mutation carriers and non-carriers. *BRCA2* mutation carriers who were estrogen receptor-positive experienced a significantly worse survival compared to those with estrogen receptor-negative disease [15]. Metcalfe et al. also reported a worse prognosis for estrogen receptor-positive versus estrogen receptor-negative *BRCA2*-associated breast cancer [16]. *BRCA1* mutation carriers with breast cancer may benefit from treatment with oophorectomy [17,18], cisplatinum [19], or olaparib [20] compared to those without a mutation. This data collectively indicates the importance role of *BRCA* mutation status with respect to treatment decisions that may impact outcome.

Here, we present a review of the literature surrounding the management of *BRCA*-associated breast cancer, with an emphasis on the prevention options that are currently available. We will provide an overview of the key findings from the literature published to date, with a focus on data from studies that are well-powered, and preferably prospective in nature. The discussion will conclude with a review of the preclinical and experimental data, pointing toward novel, non-surgical prevention options for this high-risk population.

## 2. Modifying Factors for *BRCA*-Associated Breast Cancer

We (and others) have evaluated the role of various reproductive, hormonal, and modifiable factors in the etiology of this disease [21]. To date, most of the studies have been retrospective and limited by small sample sizes, particularly in analyses stratified by *BRCA* mutation type or age at diagnosis [21,22]. Many of the risk factor associations vary by mutation type (i.e., *BRCA1* versus *BRCA2*); however, whether this is attributed to the smaller numbers of *BRCA2* mutation carriers included in most of the analyses or is a true biologic phenomenon given the differing risk profiles and pathologic features of the cancers depending on the mutated gene is unclear.

One of the strongest risk factors for both sporadic and hereditary breast cancer is family history of the disease [23]. Although Metcalfe et al. recently reported that those women with three or more first-degree relatives had significantly higher risks of developing breast cancer compared to those with no affected first-degree relative (hazard ratio (HR) = 2.29; 95% CI 1.29–4.08), the risk of breast cancer still remained substantially high among those with no family history of the disease, which was estimated at 60% by age 80 compared to 63% for those with any family history [24]. Family history is not a modifiable risk factor, but must be taken into account during risk prediction and genetic counseling.

Similar to what has been observed among women in the general population, various reproductive and hormonal factors are also associated with the risk of hereditary breast cancer. Freibel et al. have previously conducted a systematic review and meta-analysis of modifiers of cancer risk in *BRCA* mutation carriers, including studies published through 2013 [21]. In several instances, summary estimates were not reported due to the existence of overlapping studies or insufficient evidence. They reported that a later age at first birth was the only probable risk factor for *BRCA1*-associated breast cancer, with less evidence for the role of reproductive factors for women with a *BRCA2* mutation. Below is an overview of the largest studies conducted to date on various risk factors.

A later age at menarche and breastfeeding are protective for *BRCA1*-associated breast cancer, but do not appear to influence risk in women with a *BRCA2* mutation. Kotsopoulos et al. previously reported a significant 15% reduction in *BRCA1*-associated breast cancer risk for each year of menarcheal delay, and a 54% reduction in risk for women whose age at menarche was ≥15 versus ≤11 years of age (odds ratio (OR) = 0.46; 95% CI 0.30–0.69; *p*-trend = 0.002) [25]. There was an inverse relationship for women with a *BRCA2* mutation, although this did not achieve statistical significance (0.72; 95% CI 0.37–1.38; *p*-trend = 0.49). Breastfeeding is also a significant protective factor for *BRCA1* mutation carriers; each year of breastfeeding is associated with a 19% (OR = 0.81; 95% CI 0.73–0.91) reduction in risk, while two or more years was associated with a 49% reduction in risk (OR = 0.51; 95% CI 0.35–0.74) [26]. Neither history of breastfeeding nor duration of breastfeeding appear to be associated with risk among women with a *BRCA2* mutation. Age at first birth does not appear to influence risk [27], while the impact of parity or age at menopause remains unclear among carriers of either mutation [21,28].

With respect to exogenous hormones, Kotsopoulos et al. have reported a significant 40% (OR = 1.40; 0.95% CI 1.14–1.70) increased risk of early-onset (diagnosed before age 40) breast cancer among *BRCA1* mutation carriers with ever versus never oral contraceptive use in a case-control study. The risk increase was strongest for women who initiated use prior to age 20 (OR ever versus never = 1.45; 95% CI 1.20–1.75) [29]. Each additional year of oral contraceptive use prior to age 20 was associated with an 11% increased risk of early-onset breast cancer (OR = 1.11; 95% CI 1.03–1.20). This was a large, case-control study that included 2492 matched pairs of *BRCA1* mutation carriers. The risk of breast cancer for *BRCA2* mutation carriers was not evaluated, given the lack of evidence for an effect in earlier reports [22,30]. In a recent report by Schrivjer et al., the authors presented data from three types of analyses: (1) prospective, (2) left-truncated retrospective, and (3) full-cohort retrospective; however, findings were mixed and dependent on the statistical method that was used [31]. Additional prospective studies with longer follow-up periods are necessary in order to clarify the impact of oral contraceptive use on breast cancer risk, especially given its significant protective effect on ovarian cancer in *BRCA* mutation carriers, conferring a 50% reduction in risk [32].

A prospective analysis of hormone replacement therapy (HRT) after oophorectomy and breast cancer risk in *BRCA1* mutation carriers (n = 872; 92 incident cancers, and 7.6 years mean follow-up), we recently reported no relationship with the use of any type of HRT (HR = 0.97; 95% CI 0.62–1.52); however, there were differing effects by formulation type [33]. The cumulative incidence of breast cancer among women who used estrogen plus progesterone HRT was 22%, compared to 12% among women who used estrogen-alone HRT (*p*-log rank = 0.04), which was suggestive of a potential harmful effect of progesterone-containing HRT. These findings are in line with earlier randomized controlled trials of HRT versus placebo that were conducted among women in the general population [34,35].

There is some evidence for a role of modifiable lifestyle factors, including body weight and physical activity. In the first prospective report on the relationship between body size and breast cancer, which included 3734 *BRCA* mutation carriers and 338 incident breast cancers over a mean follow-up of 5.5 years, Kim et al. showed no significant impact of various anthropometric measures (i.e., height, body mass index (BMI), weight change) on subsequent risk [36]. There was some evidence of a protective role of adiposity during early adulthood (i.e., BMI at age 18 ≥22.1 kg/m^2^ versus 18.8–20.3 kg/m^2^) and postmenopausal (HR = 0.49; 95% CI 0.30–0.82; *p* = 0.006), but not premenopausal breast cancer (HR = 1.14; 95% CI 0.75–1.73; *p* = 0.56). However, due to the multiple comparisons, this requires further confirmation. In a case-control study of 443 matched pairs with detailed information on lifetime (adolescent and early adulthood) physical activity, Lammert et al. reported that moderate physical activity between ages 12–17 was associated with a 38% decreased risk of premenopausal breast cancer (OR_Q4 vs. Q1_ = 0.62; 95% CI 0.40–0.96; *p*-trend = 0.01), but was not associated with breast cancer diagnosed after menopause (OR_Q4 vs. Q1_ = 1.53; 95% CI 0.87–2.71; *p*-trend = 0.51) [37]. Kiechle et al. have begun a randomized controlled trial in *BRCA1* and *BRCA2* mutation carriers, which included structured endurance training and the Mediterranean diet; however, the primary endpoints included adherence to the dietary intervention and an improvement in body size and physical fitness [38].

A recent meta-analysis of the prospective studies conducted in the general population suggested a 13% increased risk of breast cancer (95% CI 1.09–1.17) for current smokers, compared to never smokers [39]. Regarding whether cigarette smoking is a risk factor for cancer in *BRCA* mutation carriers within this high-risk population, a number of earlier reports, including two from our group, have yielded inconclusive results, and have been limited by small sample sizes and retrospective study designs, which are vulnerable to selection bias and information bias [40,41,42,43,44,45]. In a recent prospective analysis of *BRCA* mutation carriers that included 26,711 person-years of follow-up and 700 incident cancers (428 breast cancers, 109 ovarian cancers), tobacco smoking was a risk factor for *BRCA*-associated cancer in general, including cancers of the breast and ovary [46]. Compared to never smokers, *BRCA* mutation carriers in the highest group of total pack-years (4.3–9.8) had an increased risk of developing any cancer (HR = 1.27; 95% CI 1.04–1.56), breast cancer (HR = 1.33, 95% CI 1.02–1.75), and ovarian cancer (HR = 1.68; 95% CI 1.06–2.67) [46].

Among women in the general population, alcohol consumption has been classified as a probable carcinogen for both premenopausal and postmenopausal breast cancer [47,48]. Overall, the effect size is modest, with a 20–25% increase in risk with moderate alcohol consumption (~one to two drinks/day) [49,50]. A small number of studies have reported on the relationship between alcohol intake and breast cancer among *BRCA* mutation carriers, including four case-control and two case-only studies [44,51,52,53,54,55]. Findings have been inconclusive, and these analyses have been limited by methodological issues and inadequate statistical power. In the largest prospective assessment of alcohol consumption and breast cancer risk in women with an inherited *BRCA* mutation, which included 3067 women and 259 incident cases of primary invasive breast cancer, ever or current alcohol was not associated with the risk of breast cancer [56]. The adjusted relative risks (RRs) were 1.06 (95% CI 0.78–1.44) for ever use and 1.08 (0.79–1.47) for current alcohol use, compared to non-users. For women in the highest versus lowest quintile of cumulative alcohol consumption, the RR was 0.94 (95% CI 0.63–1.40; *p*-trend = 0.65). Furthermore, there was no relationship between alcohol and risk in the analysis stratified by the timing of initiation, *BRCA* mutation type, or the age of breast cancer diagnosis. These findings suggest that unlike women in the general population, alcohol consumption does not increase the risk of breast cancer among women with a *BRCA1* or *BRCA2* mutation.

Collectively, these epidemiologic studies suggest a role of non-genetic and non-surgical risk factors in the development of *BRCA*-associated breast cancer. Although the level of risk reduction is moderate, these studies offer important insight into the underlying pathogenesis of the cancer development of mutation carriers. This is of utmost important as we attempt to identify novel targets for prevention.

## 3. Surgical Prevention

Although invasive, and potentially associated with important physiological and psychological consequences, the role of preventive surgery remains an important consideration in the management of high-risk women.

### 3.1. Bilateral Salpingo-Oophorectomy

The data from earlier studies suggested that prophylactic oophorectomy was associated with a significant 51% (95% CI 0.37–0.65) reduction in breast cancer risk for both *BRCA1* and *BRCA2* mutation carriers [57]. In contrast, two recent prospective analyses have demonstrated that oophorectomy is not associated with the risk of breast cancer among women with a *BRCA1* mutation, but may reduce the risk for *BRCA2* mutation carriers diagnosed prior to age 50, although the latter has been based on a small number of women with a *BRCA2* mutation and requires confirmation [58,59]. These divergent findings have been attributed to various methodological biases, in particular, selection bias (reviewed in [59,60,61]). The most influential biases include a role of ‘cancer-inducing bias’ and ‘immortal-time bias’, which can be overcome in well-designed prospective studies. Specifically, we and Heemskerk-Gerritsen et al. excluded women with a history of breast cancer (i.e., prevalent cancers) and initiated follow-up at the time of genetic testing. Furthermore, oophorectomy was included as a time-dependent exposure. In turn, this limited the inflated risk reduction that was seen with oophorectomy, and allowed for the accurate allocation of person-time prior to surgery. In a very recent publication, Terry et al. similarly reported no relationship between oophorectomy and breast cancer among *BRCA* mutation carriers, as well as among non-carriers based on a tertile of absolute 10-year breast cancer risk that was estimated using the BOADICEA risk prediction model [62]. Furthermore, they reported no association between oophorectomy and breast cancer risk in their analysis stratified by age at oophorectomy or HRT use in their cohort of non-carriers.

Thus, despite what was previously reported, oophorectomy likely does not impact the risk of breast cancer in this high-risk population. Nonetheless, bilateral salpingo-oophorectomy by age 35 for *BRCA1* mutation carriers and 45 for *BRCA2* mutation is strongly recommended to prevent ovarian cancer, and has also been shown to impact on survival [7,63]. In a large prospective analysis of 5783 *BRCA* mutation carriers and a mean follow-up of 5.6 years, Finch et al. estimated that preventive oophorectomy in a woman without a personal history of cancer was associated with a significant 77% reduction in all-cause mortality (95% CI 0.13–0.39; *p* < 0.001) [63]. The reduction in all-cause mortality was present in women with (HR = 0.32; 95% CI 0.26–0.39; *p* < 0.001) and without a previous diagnosis of breast cancer (HR = 0.23; 95% CI 0.13–0.39; *p* < 0.001). Although it is not likely that oophorectomy impacts breast cancer incidence, this strong protective effect on survival confirms an important role of preventive ovarian surgery for this high-risk population.

### 3.2. Bilateral Mastectomy

Preventive, bilateral mastectomy remains the most effective means to prevent *BRCA*-associated breast cancer [64,65,66]. The current National Comprehensive Cancer Network (NCCN) guidelines state that they support “discussion of the option of risk reducing mastectomy for women on a case-by-case basis”. As mentioned above, Metcalfe et al. have clearly demonstrated that the uptake of preventive surgery varies by country of residence, with higher rates reported in Canada and the United States, and lower rates in Poland and Israel [16].

In a meta-analysis of the literature, bilateral mastectomy was associated with a significant reduction in the incidence of breast cancer, but the impact on all-cause mortality did not achieve statistical significance [67]. Compared to *BRCA* mutation carriers with two breasts intact, those who had prophylactic surgery had a significantly reduced risk of developing breast cancer. The summary relative risk for the meta-analysis was 0.11 (95% CI 0.04–0.32), and was based on data from six non-overlapping studies and a total of 2555 participants. The level of risk reduction was similar in carriers of a *BRCA1* (summary RR = 0.13; 95% CI 0.02–0.94) or *BRCA2* mutation (summary RR = 0.18; 95% CI 0.07–0.47). Although based on a small number of studies (n = two) that were not statistically significant, bilateral prophylactic mastectomy was associated with a large reduction in all-cause mortality (summary RR = 0.23; 95% CI 0.05–1.02). In the only study that evaluated the impact of mastectomy on breast cancer-specific mortality, Heemskerk-Gerritsen et al. reported no significant association between mastectomy and outcome (HR = 0.29; 95% CI 0.03–2.61), although this was based on 212 women who underwent preventive surgery and a relatively short follow-up period (median 6.13 years) post-mastectomy group [68]. Given the high uptake of intensified screening with yearly magnetic resonance imaging (MRI), additional studies evaluating the long-term outcomes following either preventive mastectomy or MRI are warranted.

Women considering preventive surgery have the option to choose either a bilateral skin-sparing mastectomy, where the nipple-areolar complex is removed, or a bilateral nipple-sparing mastectomy, which preserves the nipple–areolar complex (reviewed in [69]). Reconstruction rates among *BRCA* mutation carriers is high, and may consist of immediate or delayed reconstruction with implants, the latter of which includes a subpectoral tissue expander, followed by the later replacement of the expanders with implants [70,71]. An additional option is autologous reconstruction in the form of a transverse rectus abdominus muscle flap; however, this procedure is more complex, and may be associated with a higher rate of complications [72]. There is some concern about the residual breast cancer risk with nipple-sparing mastectomy because of the remaining tissue, although emerging data suggests no excess risk with this more conservative surgery [73]. There have been no reports on the potential relationship between the type of preventive surgery and mortality. Metcalfe et al. have reported that the type of surgery and course of reconstruction may differentially impact psychosocial functioning as well as cancer-related distress and perception of breast cancer risk [74,75]. Given that the age-specific incidence of breast cancer varies based on *BRCA* mutation type, the timing of preventive surgery should be discussed among women considering this prevention option. For example, for women with a *BRCA1* mutation, the risks are highest between ages 30–50, while for women with a *BRCA2* mutation, rates peak between 40–60 years of age [1,76]. Based on a series of simulation models, Giannakeas and Narod have shown that preventive mastectomy should be considered prior to age 50 in order to maximize the mortality benefit to age 80 [76].

## 4. Chemoprevention with Selective Estrogen Receptor Modulators (SERMs)

There are several existing guidelines for the clinical management of women at a high risk of developing breast cancer (e.g., NCCN (reviewed in [65])). Currently, chemoprevention for high-risk women, including those with a five-year breast cancer risk of ≥1.7% or personal history of atypical hyperplasia or lobular carcinoma in situ, may include a selective estrogen receptor modulator (SERM) such as tamoxifen or an aromatase inhibitor such as exemestane (postmenopausal women only). The evidence supporting a role of these chemoprevention agents has been previously described [65,77,78]. Although there are also several other agents that have been proposed as potential cancer prevention agents (e.g., aspirin, metformin, bisphosphonates) (reviewed in [77]), this review will focus on those that are currently in use among women with a *BRCA* mutation. Section 5 below will introduce a more novel approach to chemoprevention with agents where there is important evidence for *BRCA1* experimental models.

Tamoxifen is a selective estrogen receptor modulator (SERM) that is used for adjuvant hormonal therapy in estrogen receptor (ER)-positive breast cancer in both premenopausal and postmenopausal women [79]. Raloxifene is also a SERM, but it is approved only for use by postmenopausal women. From 1986–1992, four randomized control trials compared tamoxifen with placebo for breast cancer prevention in healthy women at increased risk of breast cancer (reviewed in [79,80]). The women either received tamoxifen at the standard dose (20 mg per day) or placebo for at least five years. In the two largest studies (the National Surgical Adjuvant Breast and Bowel Project-P1 (NSABP-P1) and International Breast cancer Intervention Study-1 (IBIS-1)), tamoxifen reduced the incidence of breast cancer by approximately 40%, and the protective effect extended beyond the treatment period [81,82]. In a meta-analysis of the trials, tamoxifen was associated with a 33% overall reduction in risk (HR = 0.67; 95% CI 0.59–0.76) that was sustained for five to 10 years thereafter [80]. This protective effect was limited to ER-positive (HR = 0.56; 95% CI 0.47–0.67) and ductal carcinoma in situ (DCIS) (HR = 0.72; 95% CI 0.57–0.92), but was not observed for the ER-negative disease (HR = 1.13; 95% CI 0.86-1.49). Similar risk reductions were observed in trials of postmenopausal women randomized to raloxifene versus placebo (HR = 0.66; 95% CI 0.55–0.80), which was only protective for ER-positive and not ER-negative disease or DCIS. Based on the collective evidence, five years of tamoxifen use is now recommended to high-risk women in order to reduce the risk of invasive breast cancer.

Only a subgroup analysis of the NSABP trial evaluated the effect of tamoxifen on *BRCA* breast cancer risk [83]. Among the 288 incident cases, there were eight *BRCA1* and 11 *BRCA2* mutation carriers. Despite these very small sample sizes, King et al. concluded a potential reduction in *BRCA2* (risk ratio = 0.38; 95% CI 0.06–1.56) but not *BRCA1*-associated breast cancer (risk ratio = 1.67; 95% CI 0.32–10.7) with tamoxifen use. There have not been any primary prevention trials of tamoxifen (or raloxifene) conducted specifically among women with a *BRCA1* or *BRCA2* mutation.

Although not validated as chemoprevention for primary breast cancer in *BRCA* mutation carriers, tamoxifen has been shown to prevent contralateral breast cancer by up to 50% [6,84,85,86]. In a recent meta-analysis of the literature, tamoxifen use for the treatment of a first breast cancer resulted in a significant 44% risk reduction of a second breast cancer in both *BRCA1* and *BRCA2* mutation-positive patients combined (summary HR = 0.56; 95% CI 0.41–0.76) [87]. The corresponding risk estimates were 0.47 (95% CI 0.37–0.60) for *BRCA1* mutation carriers and 0.39 (95% CI 0.28–0.54) for *BRCA2* mutation carriers. A reduction in risk was reported for women with both primary ER-positive and ER-negative disease. Gronwald et al. demonstrated that one year of tamoxifen use was associated with a 63% reduction in the risk of contralateral breast cancer (95% CI 0.37–0.75; *p* = 0.003) [86]. The maximum protective effect was seen with one year of tamoxifen use.

The data thus far suggests that tamoxifen has a role in estrogen-receptor blockade and the prevention of a contralateral breast cancer, even among *BRCA1* mutation carriers who have a tendency to develop this hormone-receptor negative disease. Although the underlying mechanisms mediating the protective role of tamoxifen on contralateral breast cancer are not clear, a reduction in mammary cell proliferation [88], the number of precursor cells [89], or in mammographic density [90] have all been proposed. Significantly higher levels of circulating estradiol (and progesterone) have also been reported among premenopausal *BRCA* mutation carriers, suggesting a dysregulation in sex-hormone signaling in these high-risk women [91].

Despite a significant reduction in breast cancer incidence, these trials raised safety concerns associated with the five-year use of tamoxifen (reviewed in [78,80]). The side-effect profile of tamoxifen includes an increased risk of endometrial cancer and venous thromboembolism; however, the risks appear to be higher among postmenopausal versus premenopausal women [81,82,92,93,94,95]. Furthermore, the increase in endometrial cancer is limited to the first five years of follow-up, and dissipates after treatment is completed. All of the SERMs are associated with a significant reduction in fractures. Although raloxifene is also associated with thromboembolic events, it does not increase the risk of endometrial cancer. Tamoxifen use increased the frequency of benign gynecologic conditions and vasomotor symptom (hot flashes, vaginal discharge, and irregular vaginal bleeding) in premenopausal women [96,97] without significantly affecting the quality of life [98,99]. Segev et al. have previously reported an increased risk of endometrial cancer with tamoxifen exposure among *BRCA* mutation carriers (OR = 3.50; 95% CI 1.51–8.10) [100]. More recently, Laitman et al. have very recently published a significantly higher observed-to-expected ratio of uterine cancer among 1310 Israeli *BRCA* mutation carries compared to non-carriers (3.98; 95% CI 2.17–6.67) [101]. Together, the data suggest that hysterectomy at the time of oophorectomy may be warranted for this high-risk population.

Few women with a *BRCA* mutation opt to take tamoxifen in the preventive setting (~6%); these rates have not changed substantially since 2009 [16,102,103]. Whether this is attributed to the side effects that are related to tamoxifen or to this drug not having been validated for primary prevention remains unclear. Liede et al. recently evaluated preferences for risk-reduction among 622 *BRCA* mutation carriers [104]. In this report, they demonstrated that breast cancer risk reduction was the most important consideration of these high-risk women, and that many more women would take a chemoprevention agent compared to how many actually had taken a drug to prevent disease. These findings suggest that the uptake of chemoprevention by women with a *BRCA* mutation may be higher, given the availability of a safe and effective drug.

There is also evidence to support the use of aromatase inhibitors such as exemestane and anastrozole for the prevention of breast cancer in high-risk women; however, there have been no trials conducted among women with a *BRCA* mutation.

## 5. Future Directions: Moving Beyond Surgical Prevention with RANK-Inhibition

The receptor activator of nuclear factor κB (RANK), its cytokine ligand (RANKL), and the soluble receptor osteoprotegerin (OPG) form a pathway in the tumor necrosis factor (TNF) and TNF receptor superfamily [105,106]. RANKL can bind to both RANK and OPG; although OPG acts as a soluble decoy receptor, given its lack of a transmembrane domain or direct signaling activity, and antagonizes RANK signaling [106,107]. This pathway was first elucidated in the 1990s in the regulation of bone resorption and remodeling. The drug denosumab is a human, anti-RANKL monoclonal antibody that has been approved to treat osteoporosis and prevent skeletal damage in breast cancer patients (or those with other solid tumors) caused by bone metastases [108]. A novel role of this pathway has also been elucidated in breast development and breast carcinogenesis; where RANK and RANKL are critical for epithelial cell proliferation and survival, as well as lobuloalveolar development [109]. Various groups have demonstrated that the progesterone-mediated up-regulation of the RANK/RANKL pathway plays a critical role in mammary epithelial proliferation, mammary stem cell expansion, and carcinogenesis [89,109,110,111,112].

Two preclinical publications are of relevance for women with a *BRCA1* or *BRCA2* mutation, which demonstrated that the genetic or pharmacologic inhibition of RANKL significantly suppressed mammary tumorigenesis in *Brca1*-deficient mice [113,114]. In *Brca1* mutant mice, the loss of *RANKL* reduced mammary tumors and tumor progression, and the inhibition of RANKL prevented mammary tumor development [114]. Further, the expansion of *Brca1* mutant mammary progenitor cells was reduced with the inhibition of RANK, supporting the paracrine activity of RANKL on RANK expression in ER-negative and PR-negative cells [115,116]. Complementary to the animal data, evidence from studies using human breast cells from *BRCA1* mutation carriers support the inhibition of the RANK pathway as a novel target for prevention. Among mammary progenitor cells from *BRCA1* mutation carriers, those that were RANK-positive had significantly higher clonogenic capacity than those that were RANK-negative [113]. In three *BRCA1* mutation carriers, treatment with denosumab resulted in an inhibited activity in progenitor cells [114]. In ex vivo three-dimensional (3D) organoid models built using *BRCA1* mutated breast cells, exposure to progesterone increased ki67 expression (proliferation marker), but denosumab treatment blocked this progesterone-induced increase in ki67 [113]. Similarly, in a pilot window study of three women, pre-denosumab and post-denosumab treatment biopsies showed a marked reduction in ki67 expression post-treatment. In addition, the proliferative activity of RANK+ breast epithelial tissue from premenopausal women who were *BRCA1* mutation carriers undergoing prophylactic mastectomy was significantly higher than in premenopausal women undergoing reduction mammoplasty [113].

Along the same lines are data from studies conducted among women with a *BRCA* mutation suggesting the significant dysregulation of circulating OPG and sex hormone levels in mutation carriers [91,117,118]. Widschwendter et al. have previously reported significantly lower mean circulating levels of OPG (as well as higher progesterone) levels among premenopausal *BRCA* mutation carriers compared to non-carrier controls [91,117]. In a prospective analysis that included 206 *BRCA* mutation carriers and an average 6.5 years of follow-up, Oden et al. reported a significant inverse relationship between plasma OPG levels and breast cancer risk [118]. The 10-year cumulative incidence for women with high plasma OPG (>median) was 9% compared to 21% for women with low OPG levels. The multivariate hazard ratio associated with high versus low OPG levels 0.25 (95% CI 0.08–0.78). Although preliminary and requiring validation in a larger cohort of mutation carriers, these findings suggest that circulating OPG levels may be predictive of subsequent breast cancer risk in women with a *BRCA* mutation, and that the integration of circulating OPG levels into existing risk prediction models will potentially enable us to more accurately identify women who are at the highest risk of developing disease.

## 6. Summary

The current National Comprehensive Cancer Network (NCCN) guidelines for the management of *BRCA* mutation carriers (with respect to breast cancer risk) consists of: (1) annual mammography and MRI between the ages of 30–75 years, (2) discussion of risk-reducing mastectomy, and (3) consideration of risk-reducing agents such as tamoxifen [119]. In a recent study of 6226 *BRCA* mutation carriers, Metcalfe et al. noted that overall, 28% of women had a mastectomy and 80% were having regular breast screening, and there were significant differences in uptake by country of residence [16]. For example, rates of mastectomy were higher in Canada versus Poland (38% versus 5%). Although tamoxifen has been shown by us and by others to reduce the risk of contralateral breast cancer by about 50% in *BRCA1* mutation carriers [85,86,120], this drug has not been validated for primary prevention in this population. Furthermore, very few women choose tamoxifen to manage their cancer risk (~5%) [16,121]. In a recent Amgen-sponsored evaluation of preferences for breast cancer risk reduction among 622 *BRCA* mutation carriers (aged 25–55 years), Liede et al. found that breast cancer risk reduction was the most important consideration of these women, and that they desired a safe chemoprevention drug that was currently not available to them [104]. Given that *BRCA1* and *BRCA2* were identified more than 20 years ago, preventive mastectomy remains the gold standard, and mutation carriers have strong preferences for chemoprevention, it is timely that an effective breast cancer risk reduction option be identified.

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
