# Peer review of "BRCA Mutations and Breast Cancer Prevention"

_cancers, 2018, doi:10.3390/cancers10120524_

Reviewer 1 Report

1)actual numbers re ovarian  cancer risk from multiple studies would give 12&rate not 21%

2)same for breast risk with ranges

3) hypothesis as to why estrogen bad in face of ER negativity

4)why has literature changed re risk reduction from oophorectomy to instead a survival benefit-what was the methodologic bias that was operational

5)is the survival benefit post oophorectomy re breast cancer specific survival or is purely overall*

6) what does the author recommend to practitioners re age at mastectomy and specifically any age group who should not have it

7)tamoxifen preventing second breast cancer but not the initial does not make intuitive sense-please discuss further

8)aspirin and metformin have also been put forward as chemo prevention- any datA in BRCA

9) does the suthor believe that if the data pans out that deosumab type drugs would be used as chemoprevention

10)last paragraph in summary not add anything to paper

110\0 need paragraph on screening to be all inclusiver

Author Response

Round  2

Reviewer 1 Report

acceptable for publication with proof reading for spelling errors and missed words

Reviewer 2 Report

The authors have address my comments from prior review satisfactorily.